# Extraction of group activation features for different sleep stages from whole-brain fMRI data using Tucker decomposition

Hua-Zhe Qi
*School of Information and Communication Engineering*
*Dalian University of Technology*
Dalian, China
32209007@mail.dlut.edu.cn

Yue Han
*School of Information and Communication Engineering*
*Dalian University of Technology*
Dalian, China
h_yue@mail.dlut.edu.cn

Bin-Hua Zhao
*School of Information and Communication Engineering*
*Dalian University of Technology*
Dalian, China
zhaobh@mail.dlut.edu.cn

Qiu-Hua Lin*
*School of Information and Communication Engineering*
*Dalian University of Technology*
Dalian, China
qhlin@dlut.edu.cn

*Abstract*—**Functional magnetic resonance imaging (fMRI) provides a window for studying brain function with high spatial resolution, enabling the investigation of additional sleep-stage features beyond electroencephalography to solve the sleeping problem. However, previous studies mostly focused on analyses of fMRI data from regions of interest at single-subject level, resulting in a lack of group features. In this study, we propose to analyze whole-brain fMRI data at multi-subject level to extract group sleep-stage activation features. More precisely, a Tucker-2 decomposition algorithm is used to analyze multi-subject fMRI data collected during different sleep stages, since this algorithm simultaneously extracts group shared and individual spatial and temporal features from multi-subject fMRI data, minimizing the mixing of group and individual features. For fMRI data of different sleep stages, we extract components of interest such as the default mode network and auditory network, examine activations of group shared spatial maps, analyze frequency fluctuations of group shared time courses, and detect significant voxel-level differences in different sleep stages using individual spatial maps. We perform experiments using fMRI data from ten subjects with available non-rapid eye movement stage 1 and 2 and the wakefulness stage. The results show new findings, e.g., most components exhibit low-frequency oscillations during both wakefulness and sleep stages. However, in some networks, high-frequency signals appear in non-rapid eye movement stages, such as the default mode network and auditory network in non-rapid eye movement stage 1. Therefore, our study provides new evidence for analyzing sleep stage features.**

*Keywords—Sleep stage, fMRI, Tucker decomposition, multi-subject fMRI data*

## I. INTRODUCTION

Sleep, an essential biological function observed across various species, plays crucial roles in several vital life processes. These include facilitating learning [1][2], managing emotions [3], and promoting restorative activities [4]. The human brain activity during sleep is typically measured using scalp electroencephalography (EEG), which is considered the gold standard for sleep scoring following the rules proposed by the American Academy of Sleep Medicine in 2007 [5]. To better describe sleep, it is now commonly used to categorize sleep stages into wakefulness (W), rapid eye movement sleep, and non-rapid eye movement sleep. Non-rapid eye movement sleep can be further subdivided into non-rapid eye movement stage 1 (N1), stage 2 (N2), and stage 3 (N3) based on the prominence of slow waves and spindles [6].

However, EEG lacks both the spatial resolution and the brain coverage required for monitoring local neuronal states [7]. Functional magnetic resonance imaging (fMRI) is important to understand brain function. It provides objective, non-invasive, and high spatial resolution for the detection of localized signatures of sleep in brain hemodynamic activity [8-10]. However, previous fMRI studies on sleep mostly rely on the blood oxygen level dependent signals of voxels within regions of interest to analyze local brain activity [11-16]. These studies focus on individual subjects, resulting in a lack of group features. Group analyses of multi-subject fMRI data facilitate the extraction of brain functional components, providing a foundation for research into brain functions and clinical diagnostics [17].

Tucker decomposition (TKD), as one of the tensor decomposition algorithms, has attracted increasing attention in blind source separation of multi-subject fMRI data [18]. TKD can effectively decompose a tensor into factor matrices and a core tensor without compressing the original data [19]. From multi-subject fMRI data, TKD can extract more individual features than other tensor decomposition methods [20], in addition to shared components, so it can also minimize the mixing of group and individual features. Thus, it is promising for identifying new features for analyzing sleep stages at both group and individual levels by TKD. The sparsity-low rank-constraints TKD (slcTKD) model fully considered the characteristics of fMRI data [21], which makes full use of the tensor structure of a three-way (voxel × time × subject) multi-subject fMRI data to obtain group shared time courses and spatial maps, and the core tensor. The core tensor can be sequentially used to extract more features in individual-level, such as individual spatial maps [22].

In this study, we propose to analyze whole-brain fMRI data at multi-subject level to extract group sleep-stage activation features. We focus on four components of interest closely related to sleep: the occipital pole visual source, default mode network, auditory, and executive control network. These networks were obtained using the template obtained from the group independent component analysis of nearly 30,000 subjects by Smith et al. [23]. To extract group shared spatial components and temporal courses, the slcTKD model was applied to fMRI data from 10 patients, which included data from W, N1, and N2. Spatial activation analysis was conducted on the shared spatial maps to examine changes in spatial activation across different sleep stages. Fast Fourier Transform of the shared time courses was performed to analyze frequency variations during different sleep stages. Additionally, a paired samples t-test was conducted on individual spatial maps between different sleep stages, with a significance threshold set at $p < 0.05$.

To summarize, this research mainly consists of two parts:

- We propose to analyze different sleep stages in both group and individual levels, using the TKD model for multi-subject fMRI data. We extract group-level activation features using shared spatial maps, analyze frequency fluctuations of shared time courses, and detect voxel-level features by individual spatial maps. This allows us to investigate the spatial and frequency changes in brain activity during wakefulness and sleep.

- New temporal and spatial changes are found, e.g., high-frequency signals appear in the non-rapid eye movement stages, such as the default mode network and auditory in N1.

## II. METHODS

### A. Extraction of Shared and Individual Components

First, we aim to obtain shared spatial map (SM) and time courses (TC). For a multi-subject fMRI data $\underline{\mathbf{X}} \in \mathbb{R}^{V \times T \times K}$, where $V, T, K$ denote the number of in-brain voxels, time points, and subjects. The Tucker-2 model is built as follows:

$$\underline{\mathbf{X}} = \underline{\mathbf{G}} \times_1 \mathbf{S} \times_2 \mathbf{B} + \underline{\mathbf{E}} \tag{1}$$

where $\times_1$ and $\times_2$ denote the mode-1 and mode-2 product, $\mathbf{S} = \{\boldsymbol{s}_n\} \in \mathbb{R}^{V \times N}$ and $\mathbf{B} = \{\boldsymbol{b}_n\} \in \mathbb{R}^{T \times N}$ represent the shared SM matrix and the shared TC matrix, respectively; $\boldsymbol{s}_n \in \mathbb{R}^V$ and $\boldsymbol{b}_n \in \mathbb{R}^T$ are the spatial and temporal components obtained from shared SM and shared TC; $N$ is the order of the model. $\underline{\mathbf{G}} \in \mathbb{R}^{N \times N \times K}$ and $\underline{\mathbf{E}} \in \mathbb{R}^{V \times T \times K}$ are the core tensor and the residual tensor. In slcTKD algorithm [21], the tensors and matrices are updated as follows:

$$\min_{\mathbf{S}, \mathbf{B}, \underline{\mathbf{G}}, \underline{\mathbf{E}}} \quad ||\underline{\mathbf{X}} - \underline{\mathbf{G}} \times_1 \mathbf{S} \times_2 \mathbf{B} - \underline{\mathbf{E}}||_F^2 + \|\mathbf{S}\|_F^2 + \|\mathbf{B}\|_F^2$$
$$+ \delta \|\mathbf{S}\|_p + \lambda \|\underline{\mathbf{G}}\|_1 + \gamma \|\underline{\mathbf{E}}\|_1 \tag{2}$$

where $\|\cdot\|_F$, $\|\cdot\|_p$ and $\|\cdot\|_1$ denote low-rank $\ell_F$ constraint, $\ell_p$ sparsity constraints ($0 < p \le 1$) and $\ell_1$ sparsity constraints. The positive parameters $\delta$, $\lambda$, and $\gamma$ control the sparsity of

several regularization terms. More details about the updating rules of the shared TC matrix $\mathbf{B}$ and shared SM matrix $\mathbf{S}$ can be referred to [21].

Second, we extract individual SMs. For subject $k$, all the voxels in specific index $n$ are extracted as follows：

$$\hat{\mathbf{S}}^k = \mathbf{S} \cdot \underline{\mathbf{G}}(:,:,k) \tag{3}$$

where $k = 1, \ldots, K$, $\underline{\mathbf{G}}(:,:,k)$ is the $k$th frontal slice of tensor $\underline{\mathbf{G}}$, $\hat{\mathbf{S}}^k = \{\hat{\boldsymbol{s}}_1^k, \ldots, \hat{\boldsymbol{s}}_N^k\} \in \mathbb{R}^{V \times N}$ represents the individual SM matrix for subject $k$, $\hat{\boldsymbol{s}}_n^k$ is the spatial component extracted from individual SM matrix $\hat{\mathbf{S}}^k$, $n = 1, \ldots, N$.

### B. Component Selection

The component of interest from the shared SM matrix $\mathbf{S}$ is selected as follows:

$$n^* = arg \max_{n=1,..,N} \left( \frac{\boldsymbol{s}_n \cap \boldsymbol{s}_{ref}}{\boldsymbol{s}_{ref}} \cdot \frac{\boldsymbol{s}_n \cap \boldsymbol{s}_{ref}}{\boldsymbol{s}_n} \right) \tag{4}$$

where $\boldsymbol{s}_{ref}$ is the spatial reference; $\boldsymbol{s}_n$ denotes spatial component obtained from shared SM; "$\cap$" denotes the number of voxels activated in both $\boldsymbol{s}_n$ and $\boldsymbol{s}_{ref}$. Eq. (4) aims to select a component with more voxels inside the spatial reference network as well as less voxels outside it. For simplicity, we denote the components of interest extracted from the shared SM matrix $\mathbf{S}$ as $\boldsymbol{s}_* \in \mathbb{R}^V$. Using a similar method as described in Eq. (4), the component extracted from the shared TC matrix $\mathbf{B}$ is $\boldsymbol{b}_* \in \mathbb{R}^T$. The selected spatial map for subject k from matrix $\hat{\mathbf{S}}^k$ is denoted as $\boldsymbol{s}_*^k \in \mathbb{R}^V$.

### C. Paired Samples t-test

We denote the matrix $\hat{\mathbf{S}}_*^K = \{\boldsymbol{s}_*^1, \ldots, \boldsymbol{s}_*^K\} \in \mathbb{R}^{V \times K}$ as the SM matrix of interest for each $K$ participants, which is formed by individual SMs of interest $\boldsymbol{s}_*^k$ from a same sleep stage. The SM matrices of sleep stage W, N1, and N2 are represented by $\hat{\mathbf{S}}_{*,W}^K \in \mathbb{R}^{V \times K}$, $\hat{\mathbf{S}}_{*,N1}^K \in \mathbb{R}^{V \times K}$ and $\hat{\mathbf{S}}_{*,N2}^K \in \mathbb{R}^{V \times K}$ respectively. A paired samples $t$-test is performed between each pair of voxel vectors in different sleep stages. For example, a paired samples $t$-test between W and N2 is represented as follows:

$$\boldsymbol{t}(v) = \Phi(v) \cdot ttest\left(\hat{\mathbf{S}}_{*,W}^K, \hat{\mathbf{S}}_{*,N2}^K\right) \tag{5}$$

where $\boldsymbol{t} \in \mathbb{R}^V$ is the difference $t$-map, $ttest(\cdot)$ denotes paired samples $t$-test, and $\boldsymbol{\Phi} \in \mathbb{R}^V$ is a binary mask determined by the significant level, defined as:

$$\Phi(v) = \begin{cases} 1, & if \left|ttest\left(\hat{\mathbf{S}}_{*,W}^K, \hat{\mathbf{S}}_{*,N2}^K\right)\right| > \boldsymbol{t}_{th} \\ 0, & otherwise \end{cases} \tag{6}$$

where $\boldsymbol{t}_{th}$ is the threshold of $t$-value for paired samples $t$-test.

## III. EXPERIMENTAL METHODS

### A. fMRI Data Acquisition and Preprocessing

The fMRI data used in this study were publicly provided on the public neuroimaging repository OpenNeuro.org [24]. The

study involved 33 healthy subjects. The fMRI data were acquired using an Echo Planar Imaging sequence, with parameters as TR = 2100 ms, TE = 25 ms, flip angle = 90 degrees, slice thickness = 4 mm, number of slices = 35, FOV = 240 mm, in-plane resolution = 3 mm × 3 mm. Dataset provides the results of EEG sleep staging. Based on the results of EEG sleep staging, 10 subjects who exhibited the W, N1, and N2 sleep stages were selected. After the collection of fMRI data, several preprocessing steps were necessary to eliminate potential noise factors, using the SPM toolbox [25]. The main preprocessing steps included: (1) discarding the first ten volumes to mitigate the effects of magnetic saturation, (2) motion correction, (3) spatial normalization using the T1 structural image, and (4) spatial smoothing. The resulting fMRI data comprise 150 time points and contain 53 × 63 × 46 voxels.

### B. Experimental Process

Fig. 1 shows the flow chart of the overall experimental process. First, the experimental procedure begins with the classification of sleep stages derived from EEG signals. Subsequently, fMRI data corresponding to these stages are selected and then are subjected to preprocessing procedures. The preprocessed data are constructed into a multi-subject fMRI tensor. Second, the slcTKD method is applied to extract shared SMs and shared TCs across subjects. Individual SMs for each participant are then calculated. Third, analyses are conducted based on four networks to extract components of interest. Then, we conduct group-level analyses. For the extracted shared SMs of interest, the total number of voxels $V_{total}$ and the number of voxels inside the reference $V_{in}$ are calculated. For the shared TCs, the power spectrum is obtained by a fast Fourier transform. Finally, to capture individual-level differences, a paired samples $t$-test is conducted across different sleep stages on the individual components of interest, considering a level of $p < 0.05$ to denote statistically significant differences. Using a $t$-distribution table with 9 degrees of freedom, the critical $t$-value is 1.833.

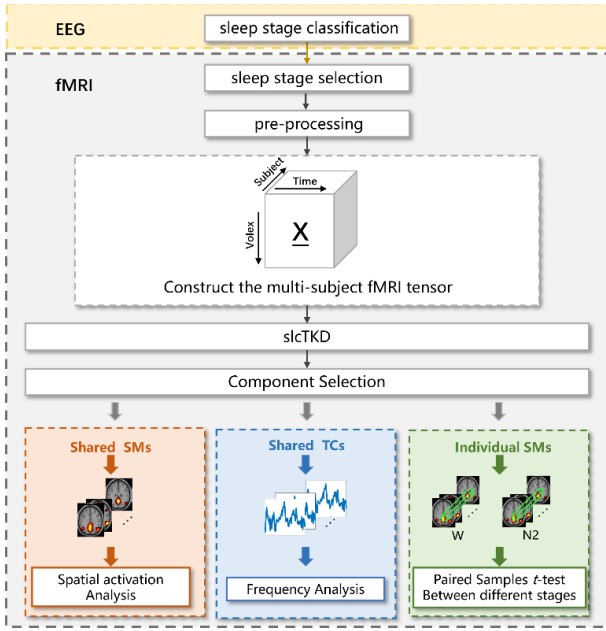

Fig. 1. The flow chart of the overall experimental process for analyzing shared spatial maps (SMs) and time courses (TCs), and individual SMs across different sleep stages.

## IV. RESULTS

### A. Group Shared Spatial Activation Maps

We focus on four components of interest closely related to sleep: the occipital pole visual source (OPVS), default mode network (DMN), auditory (AUD), and executive control network (ECN). Group shared spatial activation maps are illustrated in Fig. 2. During W, AUD and ECN are highly active. From W to N2, the shared SMs of most components become closer to the SM reference, with the values of $V_{in}$ first decreasing and then increasing, except for the DMN. Additionally, the maximum spatial activation values increase, except for the OPVS. For the OPVS, the $V_{total}$ consistently increases and $V_{in}$ reaches its maximum value of 3541 during the N2 stage, indicating that OPVS is most active during N2. The shared SMs of DMN have more activation voxels in total during the N1 and N2 stages compared to the W stage. During the N1 stage, there is a significant increase in activation in the posterior cingulate cortex. In the N2 stage, activity in the anterior cingulate cortex decreases. For the AUD, from W to N2, activation becomes increasingly concentrated in the left and right parts of the AUD network. The activation of the ECN is concentrated in the frontal and parietal cortices, and as sleep deepens, the activation becomes more concentrated in the frontal cortex.

### B. Frequency Features of Group Shared Time Courses

The power spectrum of four networks is illustrated in Fig. 3. We notice that all four components predominantly exhibit low-frequency signals (< 0.1 Hz) at each stage, with the low-frequency power being significantly greater in N2. Additionally, noticeable high-frequency signals (> 0.1 Hz) are observed in the DMN and AUD during N1.

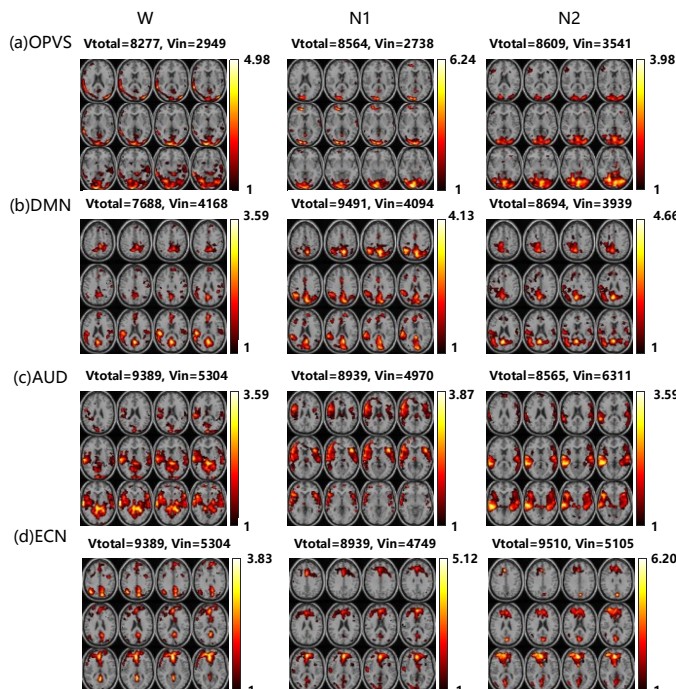

Fig. 2. Group shared spatial activation maps of four components of interest across sleep stages W, N1, and N2:(a)OPVS, (b)DMN, (c)AUD and (d)ECN.

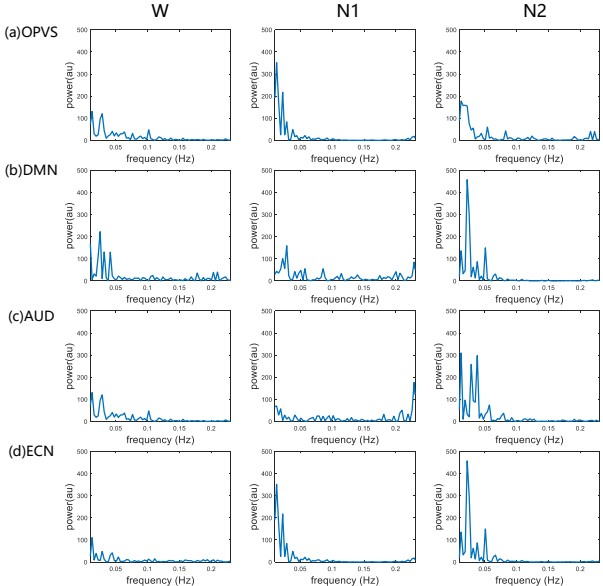

Fig. 3.  Power spectrum of four components of interest across different sleep stages (W, N1, N2): (a)OPVS, (b)DMN, (c)AUD and (d)ECN.

For DMN, in the N2 stage, the amplitude of low-frequency components increases, while high-frequency components decrease compared to the N1 stage. For the AUD, compared to the W stage, the N1 stage shows a decrease in low-frequency activity and an increase in high-frequency activity. For the ECN, during the N1 and N2 stage, there is an increase in low-frequency components, compared to the W stage.

### C.  W-N2 difference t-maps

Given that the N1 stage is a transitional stage of sleep and its differences from other sleep stages are less obvious, we present only the results of the paired samples $t$-test between the W and N2 stages. The results are shown in Fig. 4. Significant differences are primarily concentrated in the posterior occipital lobe of the brain, the anterior cingulate cortex of the DMN, and the frontal cortex of the ECN, which is consistent with the conclusions of shared SMs. The right part of auditory cortex exhibits more significant changes than the left part of auditory cortex during the transition from wakefulness to sleep.

## V.  DISCUSSION AND CONCLUSION

In this study, we utilized the slcTKD model to analyze multi-subject fMRI data, aiming to extract group activation features across different sleep stages. Our research focused on four components: OPVS, DMN, AUD, and ECN, analyzing their dynamic changes during different sleep stages. We found that each network exhibited variations in space and frequency between wakefulness and sleep.

During the transition from wakefulness to sleep, our results show that the spatial activation pattern of DMN is consistent with previous studies, with reduced activation of the anterior cingulate cortex region during N2 [26]. In addition, frequency analysis revealed a pattern consistent with previous studies: a low-frequency oscillation prominent in light sleep [7]. These findings confirm the reliability of our method.

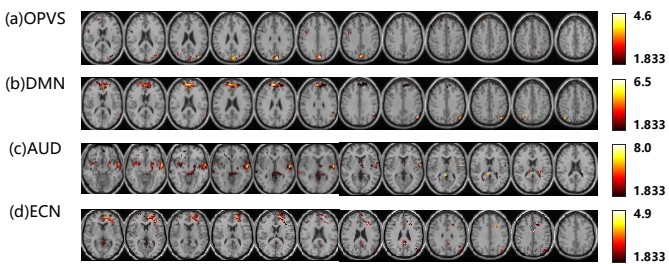

Fig. 4.  Significant difference betweenW-N2 detected using a paired samples $t$-test at $p < 0.05$ in (a)OPVS, (b)DMN, (c) AUD, and (d)ECN.

Additionally, we have made several new findings. From the results of group shared SMs, during the wakefulness stage, most networks exhibited substantial activation, with large activation areas, particularly the AUD and ECN. This suggests high brain activity in processing external stimuli and performing control tasks during wakefulness. From W to N2, the spatial activation of most components becomes more concentrated in the reference network. This suggests that as the brain transitions from wakefulness to deeper sleep stages, neural activity becomes more focused within specific functional networks. From a frequency perspective, most networks predominantly exhibit low-frequency signals during the W, N1, and N2 stages. However, the amplitude of low-frequency power is significantly higher in the N2 stage compared to the W stage. Additionally, high-frequency signals are observed in the DMN and AUD during the N1 stage. The results of the paired samples $t$-test shows the same changes as the shared SM results, indicating that the group-level changes are also consistent at the individual level. The group shared SMs indicate increased spatial activation of OPVS, AUD, and ECN components within the reference network. The results of the paired samples $t$-test also show significant changes in these regions. Additionally, the result of group shared SMs shows the anterior cingulate cortex region deactivation in the N2 stage compared to the W stage, which is also confirmed by the $t$-test result.

Our findings indicate differences in activation patterns and frequencies of the brain across different sleep stages. These insights into the dynamic changes in brain function during sleep are crucial for accurate assessment of sleep disorders and the development of targeted therapeutic strategies.

**Limitations:**

The study included only ten subjects with recorded data for the W, N1, and N2 sleep stages, lacking comprehensive analysis of the N3 and REM stages. And the physiological mechanisms underlying these changes remained unclear.

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
