# OpenReview forum: "Extraction of group activation features for different sleep stages from whole-brain fMRI data using Tucker decomposition"
_IEEE.org/ICIST/2024/Conference — IEEE ICIST 2024 Conference Submission_

### Official Review · Reviewer_k1nJ · 2024-08-25
**minor repair**

**Rating:** 8
**Confidence:** 3

**Review:**

1. When summarizing in the introduction, the author claims to have three parts, but only provides two parts. Please confirm.
2. When referencing formulas, they must be consistent, such as "Equation (4)" or "Eq. (4)".
3. I would suggest the authors polish the language of this paper carefully

---

### Official Review · Reviewer_ekKX · 2024-09-03
**Accept**

**Rating:** 7
**Confidence:** 3

**Review:**

This work is solid and recommended to be accepted

---

### Official Review · Reviewer_n1uy · 2024-09-03
**Extraction of group activation features for different sleep stages from whole-brain fMRI data using Tucker decomposition**

**Rating:** 7
**Confidence:** 4

**Review:**

The paper is well-structured and easy to follow. The introduction provides a clear background and motivation for the study. However, some figures and tables could benefit from more detailed captions to improve readability for readers who may not be familiar with the specific terminology and methods. Adding brief explanations of key terms and abbreviations in the captions would make the paper more accessible to a wider audience.

---

### Decision · Program_Chairs · 2024-09-06

Accept (Oral)